# Challenges and Pitfalls in CT-Angiography Evaluation of Carotid Bulb Stenosis: Is It Time for a Reappraisal?

**DOI:** 10.3390/life12111678

**Published:** 2022-10-22

**Authors:** Antonio Pierro, Pietro Modugno, Roberto Iezzi, Savino Cilla

**Affiliations:** 1Radiology Department, Cardarelli Regional Hospital, 86100 Campobasso, Italy; 2Vascular Surgery Unit, Gemelli Molise Hospital, 86100 Campobasso, Italy; 3Radiology Department, Fondazione Policlinico Universitario A. Gemelli-IRCCS, 00168 Rome, Italy; 4Medical Physics Unit, Gemelli Molise Hospital, 86100 Campobasso, Italy

**Keywords:** carotid stenosis, CT-angiography, NASCET, ECST

## Abstract

We aimed to perform an anatomical evaluation of the carotid bulb using CT-angiography, implement a new reliable index for carotid stenosis quantification and to assess the accuracy of relationship between NASCET and ECST methods in a large adult population. The cross-sectional areas of the healthy carotid at five levels were measured by two experienced radiologists. A regression analysis was performed in order to quantify the relationship between the areas of the carotid bulb at different carotid bulbar level. A new index (Regression indeX, RegX) for carotid stenosis quantification was proposed. Five different stenoses with different grade in three bulbar locations were simulated for all patients for a total of 1365 stenoses and were used for a direct comparison of the RegX, NASCET, and ECST methods. The results of this study demonstrated that the RegX index provided a consistent and accurate measure of carotid stenosis through the application of the ECST method, avoiding the limitations of NASCET method. Furthermore, our results strongly depart from the consolidated relationships between NASCET and ECST values used in clinical practice and reported in extensive medical literature. In particular, we highlighted that a major misdiagnosis in patient selection for CEA could be introduced because of the large underestimation of real stenosis degree provided by the NASCET method. A reappraisal of carotid stenosis patients’ work-up is evoked by the effectiveness of state-of-the-art noninvasive contemporary carotid imaging.

## 1. Introduction

Carotid artery stenosis (CAS) is a major cause of ischemic stroke, and the stenosis degree is the most important parameter in the choice of therapeutic options [1]. The benefits of performing endarterectomy in patients with symptomatic high-grade stenosis (50–99%) or in high-grade asymptomatic stenosis (60–99%) has been widely proven by the North American Symptomatic Carotid Endarterectomy Trial (NASCET) [2], the European Carotid Surgery Trial (ECST) [3], and the Asymptomatic Carotid Atherosclerosis Study (ACAS) [4,5]. Hence, a reliable determination of stenosis degree is crucial for a correct therapeutic approach.

Two different methods, NASCET and ECST, have been usually used to measure the percentage of stenosis of the internal carotid artery (ICA). For a quantitative evaluation, the degree of stenosis is measured in terms of the percentage reduction of diameter of the lumen with respect to a reference site, i.e., the supra-bulbar internal carotid artery for the NASCET method and the stenosis site for the ECST method. These different methods resulted in different degrees of stenosis, leading to potential confusion in clinical practice [6]. For example, the carotid bulb has a normal diameter larger than both ICA and CC arteries, hence the application of the NASCET method always determines an underestimation of stenosis degree with respect to ECST method. In the ECST method, the reference site coincides with the site of maximum stenosis, and therefore it should represent the most accurate method for grading the stenosis [7]. However, the extensive calcification or positive remodeling phenomena of the carotid bulb in response to atherosclerotic plaque may change the real dimensions of the reference site then reducing accuracy [8,9].

A first attempt to alleviate the ambiguities of NASCET and ECST methods was the presentation of the Common Carotid method (CC), based on the assumption of the common carotid artery as reference site [10]. Although the CC method has not yet been investigated by prospective studies with large patient populations, a few investigations reported similar evaluations of stenosis degree between CC and ECST methods [11].

Several publications focused on the determinations of mathematical relations between the various methods in order to provide a reliable comparison in stenosis degree [11,12]. Staikov et al. [11] obtained a linear regression between the three methods in which a stenosis gauged by one method can be converted into a stenosis measured by another method (e.g., 70% NASCET stenosis equals a 78% ECST stenosis). Saba et al. [12] proposed a quadratic regression curve between NASCET and ESCT (e.g., 70% NASCET stenosis equals 82% ECST stenosis). All these different relationship between the three methods may feed further confusion in clinical practice.

In order to alleviate these ambiguities, a new index called Carotid Stenosis Index (CSI), based on a fixed anatomical relationship between the CC and ICA, was proposed [13]. However, this approach has been criticized by other authors because the relationship between these vessels was found to be far from fixed, with an average CCA to ICA ratio of 1.0 with a range of 0.7 to 1.4 [10].

In this paper, we performed an anatomical study of healthy carotid arteries using CT-angiography. The normal area of the common carotid artery, carotid bulb, and internal carotid artery on cross-sectional images were evaluated in a large adult population. Based on the anatomical findings, we performed the following three tasks. First, a mathematical model was built to describe the relationship existing between different areas at different locations in the carotid bulb. Using a regression analysis, these relationships allowed to derive the cross-sectional areas of the carotid bulb at different locations in order to obtain, even at the point of maximum stenosis, the original artery dimension at that location. Therefore, this approach always allows the estimation of the carotid stenosis applying the ECST approach. Secondly, we proposed a new index for carotid stenosis quantification, termed Regression Index (RegX), and we compared this index with the current methods (NASCET and ECST) for the evaluation of different degree stenosis. Last, hundreds of simulated stenosis were obtained at each location and the relationships between NASCET and ECST were deeply investigated and compared with the consolidated ones used in clinical practice. Assuming ECST as benchmark method, the sensibility, specificity, and accuracy of the NASCET method was also evaluated.

## 2. Materials and Methods

### 2.1. Study Population

Carotid CT-angiography was obtained in 91 consecutive patients referred to our department for carotid stenosis evaluation in the period between October 2016 and October 2020. CT-angiography examination was performed when clinically indicated, usually performed when a previous Doppler Ultrasound (DUS) analysis indicated pathological stenosis or when DUS could not provide adequate information about stenosis degree, patients with hostile neck, large calcified plaques with acoustic shadowing, or high carotid bifurcation. No patients have any renal function limitations.

All patients were identified through the RIS/PACS (radiology information system/picture archiving and communication system). The patient information was anonymized to protect patient confidentiality. All patients had unilateral disease at bulb level and a contralateral healthy carotid artery without atherosclerosis. The present study dealt with the analysis of healthy carotids.

This study was conducted in accordance with the Declaration of Helsinki (as revised in 2013). We consulted extensively with the internal meeting Integrated Research Ethics Board who determined that our study did not need ethical approval, because this was a purely observational and retrospective study, focusing only on the anatomy of healthy carotids, involving non-invasive procedures, and without experimental intervention or clinical and therapeutic implications for the patient’s disease. Written informed consent was obtained from all the patients for publication of this study and any accompanying images. The safeguard of the of Belmont Report principles of non-maleficence and justice was guaranteed [14].

### 2.2. CT Angiography Protocol

CT-angiography was performed with a 128-slice scanner (Brilliance 128, Philips Healthcare, Best, The Netherlands). An 18-gauge intravenous catheter was placed in the antecubital vein; 55 mL of contrast, iomeprol 400 mg/mL (Iomeron^®^, Bracco, Milan, Italy) was infused at 4 mL/s after an initial injection delay depending on an attenuation of 140 Hounsfield units in the ascending aorta with a slice thickness of 0.9 mm. Curved multiplanar and volume rendering reconstructions were obtained by means of a dedicated computer software.

### 2.3. Carotid Artery Assessment

Two experienced radiologists performed all measurements of luminal carotid areas. Each measurement was performed three times and the final sectional areas were determined by the arithmetical mean.

The cross-sectional images of the unilateral healthy carotids were obtained on the curved multiplanar reconstruction. These images were perpendicular to the longitudinal axis of the common carotid, the carotid bulb, and the internal carotid artery. On cross-sectional images, the areas of the normal lumen were obtained at pre-established levels. The areas of the common carotid artery (a) and the internal carotid (c) were measured below the carotid bifurcation and above the bulb where the arteria walls run parallel. The areas of the carotid bulb were measured at the origin (b1), at the middle third (b2), and at the distal third (b3), as reported in Figure 1a. Inter-reader and intra-reader agreements were analyzed by calculating the intra-class correlation coefficient (ICC) and by using Cohen kappa statistics.

### 2.4. Regression Analysis

Descriptive statistics were reported using box-and-whisker plots in order to obtain a complete representation of the variable’s distributions in terms of the minimum, median, mean, interquartile ranges and outliers. A linear regression was used to model the quantitative dependence of the variables bi (the sectional area of the carotid at the i-th position of the bulb) through a linear combination of them. The resulting equations allow to extrapolate the values of the bi areas using the value of at least one of them, thus allowing to quantify the carotid stenosis in the same way used in the ECST method. Details of regression analysis are provided as supplementary data. The corresponding stenosis index was defined RegX (Regression indeX). For example, a stenosis present at b1 level could be evaluated using the effective values of the b2 area:(1)$$b_{1}^{*}=a_{1}b_{2}+e_{1}$$
(2)$$RegX=\frac{\left(b_{1}^{*}-Y\right)}{b_{1}^{*}}$$
where **Y** the residual lumen.

Statistical analyses were performed using the XLSTATSTM software (Addinsoft, New York, NY, USA).

### 2.5. Stenosis Simulation

Finally, in order to compare the degree of stenosis obtained using the RegX index with the values obtained with other methods, we simulated the presence of several stenoses with different grade for all patients at different bulb levels. In particular, we hypothesized the existence of stenoses at b1, b2, and b3 levels with a residual patent lumen equal to 2, 4, 6, 8 and 10 mm^2^, for a total of 455 stenosis at each position. In this simulation analysis, ECST provides the real stenosis degree because the value of the normal healthy carotid area is real so that ECST was considered as a benchmark. Statistical comparisons of data were performed by a Kruskal–Wallis analysis of variance (ANOVA). The Bonferroni–Dunn post-hoc non-parametric test was run to correct for multiple comparisons, with adjusted *p*-values at 0.083 indicating statistical significance.

## 3. Results

### 3.1. Descriptive Statistics

Median age of the patients was 71 years (min: 43–max: 85) with 48 (52.7%) males and 43 (47.3%) females. The measurements of carotid areas at the positions described in Figure 1a are graphically represented as box-plots in Figure 1b and reported in Table 1.

Carotid cross-sectional areas in the five locations of measurements were found widely variable, but all patients presented the same morphology of reversed truncated cone shape of the carotid bulb, with b1 > b2 > b3 (Figure 1b).

The inter-reader agreement was excellent in defining the values of cross-sectional areas at all three levels of carotid bulbs with k-value equal to 0.93.

### 3.2. Regression Analysis

The regression analysis of the anatomical data of the healthy carotid arteries allowed to examine the relationship between the dimensions of the carotid bulb at pre-established points (b1, b2, and b3) and to estimate the expectation value of the cross-sectional area in one location when the others take on a given set of values.

Figure 2 shows the linear regression between b1 and b2 variables with (a) the regression line and the confidence intervals on mean and on values, (b) the plot or the normalized residuals, and (c) the plot of the Cook’s distance. A similar behavior was found for the other correlations.

Table 2 reports the equations obtained from the regression analysis that allow the prediction of the cross-sectional area in the stenosis location when the other location areas are measurable.

The regression analysis reported a very good performance in terms of R^2^ scores and normal distribution of residuals. For example, looking at the R^2^ coefficient for the relation between b1 and b2, 86.1% of the variability of the b1 variable is explained by the b2 variable. Similarly, the R^2^ coefficients for the relations b2–b3 and b1–b3 were found 83.9% and 73.0%, respectively. With respect to Cook’s distance, a metric able to detect observations that strongly influence fitted values of the model, all values were found to be less than 0.045 for the three regression models, suggesting that no points negatively affected our regression models. The histogram of the residuals, as shown in Figure 2b, enabled us to visualize the residuals that are out of the range [−2, 2]. Values outside this interval are potential outliers and may suggest that the normality assumption is wrong. In this analysis, a maximum of three residuals was found out of range for each regression analysis, meaning that, given the assumptions of the linear regression model, residuals are normally distributed (i.e., 95% of the residuals are in the interval [−2, 2]).

### 3.3. Stenosis Simulation and Comparison between NASCET and ECST

Five different stenoses were simulated at b1, b2, and b3 locations for all patients. All stenoses were obtained considering a patent residual lumen ranging from 10 mm^2^ (mild stenosis) to 2 mm^2^ (high-grade stenosis) with 2 mm^2^ increments. This means that 455 simulated stenoses were obtained at each location. Stenosis degree was quantified using the ECST and NASCET methods commonly used in literature and the RegX.

Table 3 shows the overall results, highlighting the significant agreement of the RegX with the true ECST values, for each location and degree of stenosis (*p >* 0.85). As expected, the NASCET index always underestimates the degree of stenosis with respect to ECST, in any location and for any degree of stenosis (p < 0.001). However, the relationships between NASCET and ECST values are in agreement with literature data only in b3 (i.e., a NASCET value of 69% corresponds to a ECST value of 82%). In b1 and b2 locations, the NASCET indexes significantly depart from expected data with a clear underestimation of stenosis degree. In particular, a NASCET value of 69% correspond to ECST values of 90% and 87% in b1 and b2 locations, respectively.

As example, Figure 3 shows the distributions of the four stenosis degree indexes for all stenosis in the b1 location.

Figure 4 reports with great details the relations between ECST and NASCET results for all the simulated stenosis as a scatter plot.

Assuming a 70% value of carotid stenosis as the cut-off for benefit of carotid endarterectomy, the sensibility, specificity, and accuracy of the NASCET method vary considerably at different levels of the carotid bulb.

In b1, there are no true negatives but only false negatives for NASCET less than 70%. Therefore, if the stenosis is in b1, the NASCET method has a sensibility equal to 56%, which means that 44% of severe stenoses (according to ECST above 82%) are not identified. Instead, the specificity is 100%, i.e., all healthy patients are correctly identified. In b1, the accuracy of the NASCET method is 60.4%.

Similarly, if the stenosis is in b2, the NASCET method has a sensibility of 65.9%, which means that 34.1% of severe stenosis (according to ECST above 82%) are not identified. Instead, the specificity is 99% and therefore almost all healthy patients are correctly identified. In b2, the accuracy of the NASCET method is 73.4%.

Finally, if the stenosis is in b3, the sensitivity is 91.3% and the specificity is 90.7% with an accuracy of 91%; these values are comparable to what is reported in the literature for the relationship between NASCET and ECST.

## 4. Discussion

In the present paper, we performed an anatomical CT-angiography evaluation of the carotid bulb morphology for a large adult population. Then, a regression analysis allowed to examine the relationship between the dimensions of the carotid bulb at pre-established points and to estimate the expectation value of the cross-sectional area in one location when the others take on a given set of values. In this way, we were able to always apply the ECST method to estimate carotid artery stenosis. The obtained equations are presented in Table 2 and the goodness of regression analysis allowed a reliable prediction of the cross-sectional area of the carotid bulb in one location (e.g., where there is a stenosis) when the other location areas are measurable. A new index called RegX, able to provide a reliable evaluation of the carotid bulb stenosis within the ESCT strategy, was then proposed. Moreover, we performed a simulation test based on five different stenoses with different grade from mild to very high, in various bulbar locations for all patients for a total of 1365 simulated stenoses. Comparing the RegX with the NASCET and ECST methods, it emerged that the latter provides stenosis indexes statistically not different from the true ECST ones, at each bulbar level and for each grade. In particular, mean RegX values were always found in agreement with the true mean ECST ones within only 1%. With this method, regardless of any morphological alterations of carotid bulb induced by the presence of plaque, the RegX index can be always applied, avoiding the risk of overestimating or underestimating the bulbar stenosis degree. As expected, our data also showed that the major differences between the NASCET and ECST methods are present for low stenosis degree, whereas for high stenosis degree the difference is reduced, as reported in the literature [12].

A relevant finding is the different correlation of NASCET values with the ECST ones, with respect to literature data. For example, we found that a NASCET stenosis of 48% corresponds to ECST stenosis of 83% in b1, 79% in b2, and 69% in b3. Similarly, a NASCET stenosis of 69% corresponds to ECST stenosis of 90% in b1, 87% in b2, and 82% in b3.

Currently, most guidelines suggest the use of the NASCET method for the evaluation of carotid stenoses [15]. In particular, the societal guidelines recommend carotid endarterectomy (CEA) for severe (≥70% NASCET) symptomatic carotid stenosis if an operative stroke/death rate of <6% can be maintained. Moreover, although the benefit is less evident, most guidelines also recommend consideration of CEA for 50% to 69% symptomatic stenosis [16]. CEA is the first-line treatment also for asymptomatic patients with stenosis of 60% to 99% in highly selected patients but the perioperative risk of stroke and death in asymptomatic patients must be <3% and the patient must have a 3- to 5-year life expectancy to ensure real benefit [5]. As previously stated, stenosis values of 50% and 70% evaluated with the NASCET method have been shown to be equivalent to 65% and 82% for the ECST method [6,10,11,17].

Our results indicate a large deviation from these equivalences when the stenosis is located at b1 and b2 level, where for example, NASCET stenosis of 48% corresponds to an ECST stenosis of 83% and 79% in b1 and b2, respectively. This means that, in asymptomatic patients, the application of the NASCET method could significantly underestimate the carotid stenosis, precluding the use of CEA and consequently increasing the risk of stroke for those patients. On the basis of these results, it seems reasonable to still adopt the NASCET method only at the b3 level, and that the ECST method should be used when stenosis is located at the b1 and b2 bulbar levels, where the underestimation of stenosis degree in not negligible.

At the moment, the clinical implications of our findings are difficult to assess. We are conscious that the benefit from endarterectomy depends not only on the degree of carotid stenosis, but also on several other factors, including morphologic characteristics and composition of carotid plaques [18,19,20,21]. We are aware that the goal of estimating the degree of carotid stenosis should be oriented toward the best clinical effect rather than the best anatomical excellence. For example, surgery is ineffective in the extreme scenarios of carotid pathology, i.e., in patients with near occlusion or severe stenosis with narrowing of the ICA. On the other hand, it is widely recognized that endarterectomy is of some benefit for 50% to 69% symptomatic stenosis and highly beneficial for 70% to 99% stenosis without near-occlusion.

Nonetheless, in clinical practice, the steadfastness to perform CEA is still based on the degree of carotid stenosis and the presence of relevant symptoms and as radiologists, we are required to accurately quantify the percentage of carotid stenosis. We are also aware that the NASCET method is used worldwide in clinical practice. However, our data suggest that the NASCET’s sensibility at sites b1 and b2 is very low (equal to 56.0% and 65.9%, respectively), so we cannot ignore that many critically ill patients may be missed when this method is applied. Our data clearly have only a pure anatomical meaning at this time. However, we cannot escape from the main suggestion that emerged from this study: does the application of the NASCET method using actual advanced imaging (CT- and MR-angiography) preclude the identification of a large number of patients who should be treated by endarterectomy? We believe that focused studies on this subject are necessary to answer this question.

We are fully aware that there is a reluctance to abandon the NASCET method as NASCET outcome results are considered a “dogma” of stroke prevention management. However, we cannot neglect the pitfalls of NASCET measurement [22], including (a) the lack of compliance to the specific details of how NASCET used its method, (b) the lack of assessment for near occlusion, and (c) the use of distal ICA diameter as denominator to calculate the % stenosis. The carotid bulb is an anatomic aberration, being an unusually dilated part of an artery, and it can have nearly twice the diameter of the carotid artery beyond the bulb [23]. Therefore, application of the NASCET methodology to an anatomic aberration of the carotid bulb may result in an underestimation of the stenosis degree. On the other hand, in the presence of an atherosclerotic artery, the application of the ECST method could cause an overestimation of the percentage of stenosis as a consequence of the measurement method itself. In fact, ECST compares the ICA stenosis diameter with a supposed outline of carotid and this could inevitably include the positive remodeling. However, considering the anatomic aberration of the bulb and its greater amplitude than the downstream ICA, from an anatomical purism point of view, it is more logical and correct to use the ECST method for estimating the % of stenosis.

Nevertheless, the positive or even negative remodeling that inevitably accompanies the carotid plaque will also make the ECST method fallacious. Recently, considering the high-quality anatomical performances of CTA imaging, a few authors suggested the quantification of carotid stenosis merely by measuring the mm diameter of the residual lumen, removing any reference to the denominator [22,24].

The introduction of Reg X allows to overcome the critical issues of the denominator present in both NASCET and ECST methodologies. In particular, the RegX always allows the use of a denominator “stripped” of all the uncertainties deriving from the anatomical metamorphosis generated by the carotid plaque.

A few limitations must be recognized. First, it must be highlighted that in our study, the age of the population ranged between 43 to 85 years. It has been demonstrated that the geometry of normal carotid artery changes during aging (with increasing bulb diameter), usually due to the degradation and fragmentation of intramural elastin [25]. Therefore, the findings of the present study are preferentially applicable to patients over 43 years of age. Considering that the size of the carotid bulb could suffer variations among race-ethnic groups [26], another limitation of our study may be the origin of all patients from the same geographical region of central Italy.

It must be also highlighted that the evaluation of carotid stenosis was based on the measurement of cross-sectional area rather than on narrowest artery diameter. In particular, this choice is strongly recommended when using CT-angiography and it is supported by several studies reporting that diameter-based measurements significantly underestimated the degree of carotid stenosis [27,28], particularly in the setting of an irregularly shaped lumen [29].

Last, it must be underlined that the proposed RegX index cannot be applied to bulbar stenosis involving the entire length of the bulb. However, generally, the majority of carotid plaques do not affect the entire length of the carotid bulb, but the most frequent site of stenosis is the origin of the ICA [3]. However, if this occurs, it is advisable to use the CC method, because CC and ECST methods grade the stenosis more similarly.

## 5. Conclusions

The results of this study demonstrated that the RegX index provided a consistent and accurate measure of carotid stenosis through the application of the ECST method, avoiding the limitations of the NASCET method. Furthermore, using CT-angiography, our results strongly depart from the consolidated relationships between NASCET and ECST values used in clinical practice and reported in extensive medical literature. In particular, we highlighted that a major misdiagnosis in patient selection for CEA could be introduced because of the large underestimation of real stenosis degree provided by the NASCET method. A reappraisal of carotid stenosis patients’ work-up is evoked by the effectiveness of state-of-the-art noninvasive contemporary carotid imaging.

## Figures and Tables

**Figure 1 life-12-01678-f001:**
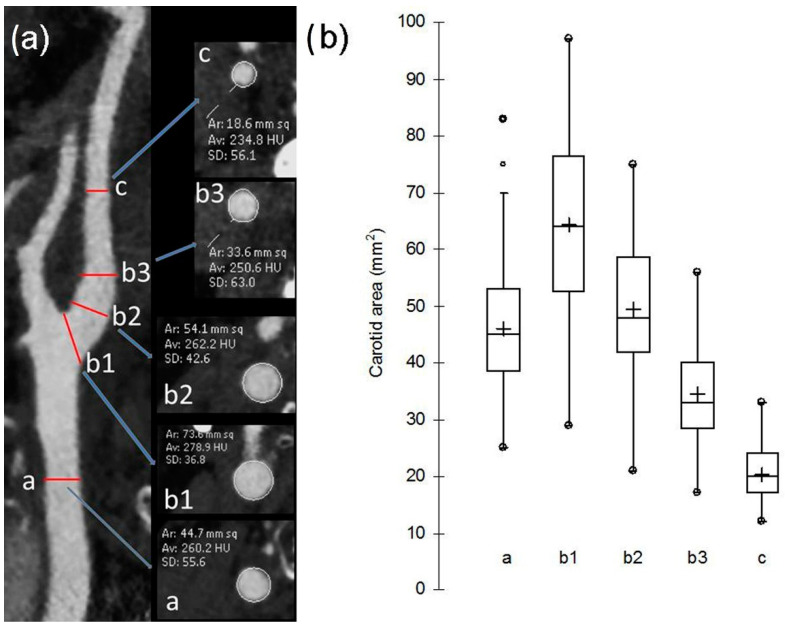
(**a**) Measurements of luminal carotid areas. The areas of the common carotid artery (a) and the internal carotid (c) were measured at 2 cm below the carotid bifurcation and 2 cm above the bulb, respectively. The areas of the carotid bulb were measured at the origin (b1), at the middle third (b2), and at the distal third (b3); (**b**) box-and whisker plots of measured carotid cross-sectional areas in the five locations shown in (**a**), reporting the median values and the interquartile range (where 50% of the data are found). Mean values are shown as crosses.

**Figure 2 life-12-01678-f002:**
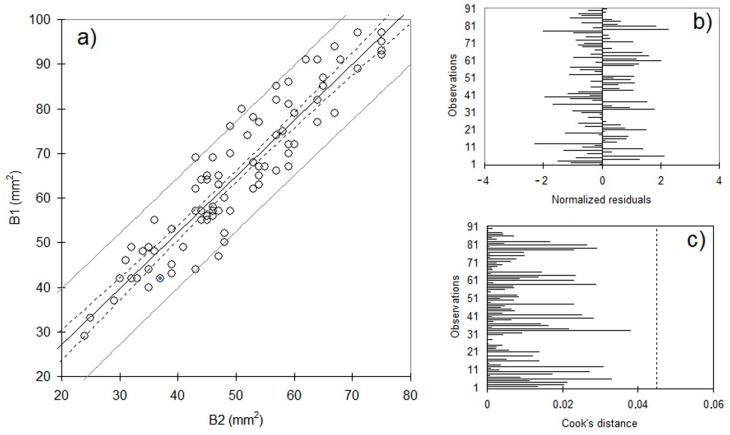
Linear regression between b1 and b2 variables with (**a**) the regression line and the confidence intervals on mean and on values, (**b**) the plot or the normalized residuals, and (**c**) the plot of the Cook’s distance.

**Figure 3 life-12-01678-f003:**
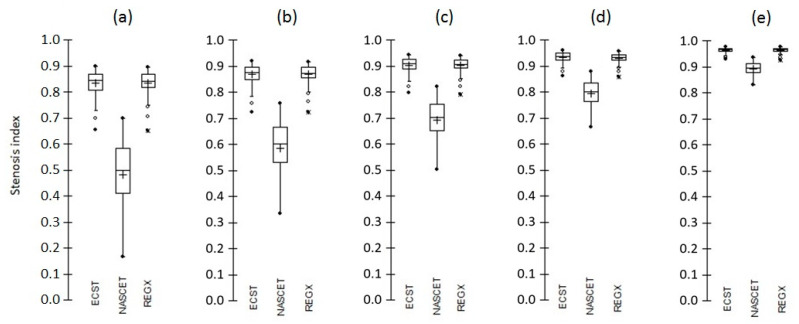
Box-plots of the distributions of the NASCET, ECST, and ReGX stenosis degree indexes for all simulated stenosis in the b1 location obtained with a residual lumen ranging of (**a**) 10 mm^2^, (**b**) 8 mm^2^, (**c**) 6 mm^2^, (**d**) 4 mm^2^ and (**e**) 2 mm^2^.

**Figure 4 life-12-01678-f004:**
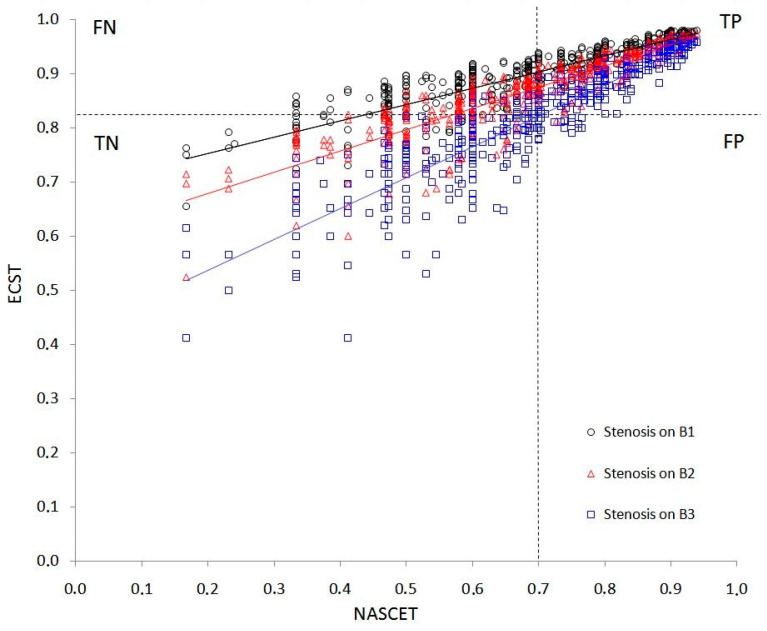
Correlation scatter plot of ECST and NASCET values in b1 (black), b2 (red), and b3 (blue) locations. An example of diagnostic test was reported considering the current equivalence relation between NASCET (70%) and ECST (82%) stenosis degree. TP (True Positives), FP (False Positives), FN (False Negatives), and TN (True Negatives).

**Table 1 life-12-01678-t001:** Carotid cross-sectional areas in the five locations of measurements in terms of mean and standard deviations (SD).

	Mean (mm^2^)	SD (mm^2^)
a	45.9	10.8
b1	64.3	16.4
b2	49.5	12.0
b3	34.5	8.7
c	20.3	4.5

**Table 2 life-12-01678-t002:** Results of the regression analysis. The equations provide the prediction of the cross-sectional area of the carotid bulb in one location (e.g., where there is a stenosis) when the other location areas are measurable. R^2^, Dw, and Dm represent the regression coefficient, residual autocorrelation parameters, and the Cook’s distances.

R^2^	D_W_	D_m_	Equations
0.861	2.147	0.009	b1 = 1.872 + 1.259 × b2
			b2 = 5.603 + 0.684 × b1
0.839	1.892	0.010	b2 = 5.837 + 1.269 × b3
			b3 = 1.693 + 0.661 × b2
0.730	1.949	0.009	b1 = 8.957 + 1.605 × b3 b3 = 5.259 + 0.455 × b1

**Table 3 life-12-01678-t003:** Overall results of simulated stenosis degree in b1, b2, and b3 evaluated using the two methods commonly used in literature (ECST and NASCET) and the RegX. Five different stenoses were simulated at b1, b2, and b3 locations considering a patent residual lumen ranging from 10 mm^2^ (mild stenosis) to 2 mm^2^ (high-grade stenosis) with 2 mm^2^ increments.

		ECST		NASCET		RegX		*p*-Values	
		Mean	SD	Mean	SD	Mean	SD	ECST vs. NASCET	ECST vs. RegX
	Residual lumen (mm^2^)								
Stenosis on B1	10	0.833	0.047	0.483	0.119	0.835	0.044	<0.0001	0.838
	8	0.867	0.037	0.586	0.096	0.868	0.035	<0.0001	0.877
	6	0.900	0.028	0.690	0.072	0.901	0.026	<0.0001	0.877
	4	0.933	0.019	0.793	0.048	0.934	0.018	<0.0001	0.877
	2	0.967	0.009	0.897	0.024	0.967	0.009	<0.0001	0.877
	Residual lumen (mm^2^)								
Stenosis on B2	10	0.784	0.061	0.483	0.119	0.787	0.051	<0.0001	0.984
	8	0.827	0.049	0.586	0.096	0.830	0.041	<0.0001	0.984
	6	0.870	0.036	0.690	0.072	0.872	0.031	<0.0001	0.984
	4	0.914	0.024	0.793	0.048	0.915	0.020	<0.0001	0.984
	2	0.957	0.012	0.897	0.024	0.957	0.010	<0.0001	0.984
	Residual lumen (mm^2^)								
Stenosis on B3	10	0.690	0.085	0.483	0.119	0.696	0.069	<0.0001	0.900
	8	0.752	0.068	0.586	0.095	0.757	0.055	<0.0001	0.892
	6	0.815	0.050	0.686	0.079	0.818	0.041	<0.0001	0.968
	4	0.877	0.033	0.792	0.049	0.878	0.028	<0.0001	0.909
	2	0.938	0.017	0.897	0.024	0.939	0.014	<0.0001	0.892

## Data Availability

The data presented in this study are available on request from the corresponding author.

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
