# Peer review of "Challenges and Pitfalls in CT-Angiography Evaluation of Carotid Bulb Stenosis: Is It Time for a Reappraisal?"

_life, 2022, doi:10.3390/life12111678_

Round 1

Reviewer 1 Report

This study compares the standard NASCET and ECST methods with the regression index (RegX) using contrast-enhanced computed tomography proposed by the authors in this study for the evaluation of stenosis in carotid artery stenosis. The authors suggest that the NASCET method is inadequate in assessing the degree of stenosis and may miss patients suitable for treatment.

#1 Although a very detailed study, it seems to indicate that RegX, which dares to use contrast-enhanced CT examinations with contrast agents, was similar to the ECST method. The authors should perform some kind of test to see if RegX is similar or superior to the ECST method and show its usefulness or merit in the discussion.

#2 Did the subject patients have any renal function limitations despite the use of contrast media? If so, please indicate if there are any renal function limitations.

Author Response

Reviewer 1:

This study compares the standard NASCET and ECST methods with the regression index (RegX) using contrast-enhanced computed tomography proposed by the authors in this study for the evaluation of stenosis in carotid artery stenosis. The authors suggest that the NASCET method is inadequate in assessing the degree of stenosis and may miss patients suitable for treatment.

We would like to thank the editor and reviewers for their valuable time and efforts in reviewing our manuscript. The comments are very thoughtful and insight in improving the quality of our manuscript. The point-by-point responses are presented below and are also reflected in the manuscript. All the changes were marked with blue font in the revised manuscript.

The itemized responses to reviewers are presented below.

#1 Although a very detailed study, it seems to indicate that RegX, which dares to use contrast-enhanced CT examinations with contrast agents, was similar to the ECST method. The authors should perform some kind of test to see if RegX is similar or superior to the ECST method and show its usefulness or merit in the discussion.

Thanks for this suggestion that allow us to better explain the aim of our paper. In the clinical practice, two major methods have been introduced to quantify carotid stenosis: the NASCET method and the ECST method.

The degree of stenosis is measured in terms of the percentage reduction of diameter of the lumen with respect to a reference site, i.e. the supra-bulbar internal carotid artery for the NASCET method and the stenosis site for the ECST method. These different methods resulted in different degree of stenosis, leading to potential confusion in clinical practice. For example, the carotid bulb has a normal diameter larger than both ICA and CC arteries, hence the application of NASCET method always determines an underestimation of stenosis degree with respect to ECST method. In ECST method the reference site coincides with the site of maximum stenosis, and therefore it should represent the most accurate method for grading the stenosis. However, the extensive calcification or positive remodeling phenomena of the carotid bulb in response to atherosclerotic plaque may change the real dimensions of the reference site then reducing accuracy.

In this paper, we introduced a new index called RegX conceptually similar to the ECST index. The potential advantage of RegX with respect to ECST is that now the RegX index uses the original dimension of the carotid (as obtained by regression modeling) and not the misrepresented dimensions caused by the remodeling of plaque. With this approach, the degree of stenosis in the point of interest is real, so that it is not influenced by the distorsions of vessel due to presence of the plaque (remodeling).  

All our measurements were obtained on healthy arteries, without remodeling phenomena. To test the agreement of RegX with ECST method (for healthy carotid) we simulated the presence of several stenosis with different grade for all patients at different bulb levels. In particular, we hypothesized the existence of stenosis at three different levels with a residual patent lumen ranging from 2 to 10 mm2, for a total of 455 stenosis at each position. In this test analysis, ECST provides the real stenosis degree because the value of the normal healthy carotid area is real so that ECST was considered as a benchmark. Statistical comparisons of data were performed by a Kruskal-Wallis analysis of variance (ANOVA). The Bonferroni-Dunn post-hoc non-parametric test was run to correct for multiple comparisons, with adjusted p-values at 0.083 indicating statistical significance.  The results obtained from this test reported that that the RegX index provides stenosis indexes statistically not different from the true ECST ones, at each bulbar level and for each grade. In particular, mean RegX values were always found in agreement with the true mean ECST ones within only 1%.

Therefore, with this method, regardless of any morphological alterations of carotid bulb induced by the presence of plaque, the RegX index can be always applied, avoiding the risk of overestimating or underestimating the bulbar stenosis degree. In addition, we reported that the NASCET index always underestimates the degree of stenosis with respect to ECST, in any location and for any degree of stenosis.

This explanation was now better discussed in the discussion section (from line 321 to 332).

#2 Did the subject patients have any renal function limitations despite the use of contrast media? If so, please indicate if there are any renal function limitations.

Thanks for this suggestion. In this paper, all CT-angiography examinations were performed when clinically indicated (i.e. when a previous Doppler Ultrasound (DUS) analysis indicated pathological stenosis or when DUS could not provide adequate information about stenosis degree, patients with hostile neck, large calcified plaques with acoustic shadowing or high carotid bifurcation). No patients have any renal function limitations. This was now inserted in material and methods section (Study population).

Reviewer 2 Report

Does this technique change the indications for carotid intervention since most recommendations based on NASCET criteria?

Does your technique invalidate the results of the NASCET study?

How does using your data differ from just using the ECST criteria?

Author Response

Reviewer 2:

We would like to thank the editor and reviewers for their valuable time and efforts in reviewing our manuscript. The comments are very thoughtful and insight in improving the quality of our manuscript. The point-by-point responses are presented below and are also reflected in the manuscript. All the changes were marked with blue font in the revised manuscript.

The itemized responses to reviewers are presented below.

Does this technique change the indications for carotid intervention since most recommendations based on NASCET criteria?

Thanks for this question. We are aware that the clinical implications of our findings are difficult to assess. NASCET had a worldwide widespread adoption because of its major advantage: Nascet approach offers to the clinic the choice of the best therapeutic option. Despite its wide diffusion in the last decades, as early as the mid-1990s, many authors questioned if the NASCET method was reliable outside of clinical trials (Gagne PJ, et al. J Vasc Surg. 1996 Sep; 24 (3): 449-55;  Fox AJ, et al. Radiology. 2009 253 (2): 574-5). These authors highlighted some pitfalls of the NASCET measurements, including a) the lack of compliance to the specific details of how NASCET used its method, (b) the lack of assess-ment for near occlusion and (c) the use of distal ICA diameter as denominator to calculate the % stenosis. Moreover, the carotid bulb is an anatomic aberration, being an unusually dilated part of an artery and it can have nearly twice the diameter of the carotid artery beyond the bulb. Then, the application of the NASCET methodology to an anatomic aberration of the carotid bulb may result in an underestimation of the stenosis degree.

We are fully aware that there is a reluctance to abandon the NASCET method as NASCET outcome results are considered a “dogma” of stroke prevention management. However, at the moment, the aim of this paper was not to suggest a clinical alternative to the Nascet approach in the clinical routine. This is an ambitious goal that need a reliable validation in a clinical trial. Based on our results, we rather highlighted the need for a reappraisal of the NASCET strategy. The pitfalls of stenosis quantification and differences between methods are problems of the ratio’s denominator, and may potentially be resolved by the use of real carotid diameter for stenosis quantification on CTA, as the proposed RegX.

Our data suggest that the NASCET's sensibility can be very low (about 60%), so we cannot ignore that many critically ill patients may be missed when NASCET method is applied. Nowadays, our data clearly have only a pure anatomical meaning at this time. However, we cannot escape from the main suggestion that emerged from this study: do the application of NASCET method using actual advanced imaging (CT- and MR-angiography) preclude the identification of a large number of pa-tients who should be treated by endarterectomy? We believe that focused studies on this subject are necessary to answer this question.

These considerations are already discussed in the discussion section.

Does your technique invalidate the results of the NASCET study?

Thanks for your suggestion. As previously discussed, the present study has not been clinically validated and, at the moment, our results are based on anatomy and mathematical regression. However, we believe that our results raise non-trivial questions that should be further investigated with focused clinical trials.

How does using your data differ from just using the ECST criteria?

One of the pitfalls of the ECST criterion is the presence of remodeling phenomena that affects the real diameter of the carotid at the measurement point. This can translate in an overestimation of the stenosis degree. On the contrary, the RegX strategy is based on the actual size of the carotid, as determined by regression analysis, which no longer considers the contribution of the positive or negative remodeling phenomena.      

Round 2

Reviewer 2 Report

Changes are satisfactory